# Neurotrophins as Potential Biomarkers for Active Disease and Poor Outcome in Pediatric Acute Lymphoblastic Leukemia

**DOI:** 10.3390/cancers17101623

**Published:** 2025-05-11

**Authors:** Karine Pereira de Andrade, Gustavo Lovatto Michaelsen, Lívia Fratini Dutra, Rebeca Ferreira Marques, Daniela Elaine Roth Benincasa, Júlia Plentz Portich, Jiseh Fagundes Loss, Lauro José Gregianin, André Tesainer Brunetto, Marialva Sinigaglia, Rafael Roesler, Mariane da Cunha Jaeger, Marcelo Land, Caroline Brunetto de Farias

**Affiliations:** 1Cancer and Neurobiology Laboratory, Experimental Research Center, Porto Alegre’s Clinical Hospital, Federal University of Rio Grande do Sul, Porto Alegre 90035-903, RS, Brazil; 2Postgraduate Program in Pharmacology and Therapeutics, Institute for Basic Health Sciences, Federal University of Rio Grande do Sul, Porto Alegre 90050-170, RS, Brazil; 3Children’s Cancer Institute, Porto Alegre 90620-110, RS, Brazil; 4Pediatric Oncology Service, Porto Alegre’s Clinical Hospital, Federal University of Rio Grande do Sul, Porto Alegre 90035-903, RS, Brazil; 5Pediatric Oncology Service, Conceição Children Hospital, Porto Alegre 91350-250, RS, Brazil; 6Hematology and Hemotherapy Service, Porto Alegre’s Clinical Hospital, Federal University of Rio Grande do Sul, Porto Alegre 90035-903, RS, Brazil; 7National Institute of Science and Technology in Childhood Cancer Biology and Pediatric Oncology—INCT BioOncoPed, Porto Alegre 90050-170, RS, Brazil; 8Department of Pharmacology, Institute for Basic Health Sciences, Federal University of Rio Grande do Sul, Porto Alegre 90050-170, RS, Brazil; 9Department of Pediatrics, Faculty of Medicine, Federal University of Rio de Janeiro, Rio de Janeiro 21941-912, RJ, Brazil; 10Martagão Gesteira Childcare and Pediatrics Institute, Federal University of Rio de Janeiro, Rio de Janeiro 21941-912, RJ, Brazil

**Keywords:** neurotrophins, childhood leukemia, prognosis, BDNF, ALL

## Abstract

Neurotrophins are essential growth factors involved in cellular development and survival. Their role in Acute Lymphoblastic Leukemia (ALL) remains to be fully elucidated. This study evaluates neurotrophin levels in pediatric ALL patients and investigates their relationship with disease phases and outcomes. Our findings suggest that decreased neurotrophin levels at diagnosis correlate with active disease and worse outcomes, indicating their potential as biomarkers when assessing ALL in children.

## 1. Introduction

Neurotrophins (NTs) are a family of growth factors known to be involved in neuronal development, survival, and plasticity [1]. In recent years, the roles of these proteins in the nervous system have been increasingly investigated. NTs are secreted by different tissues and cells, including the immune system [2], and are responsible for their development and maintenance. The NT family includes nerve growth factor (NGF), brain-derived neurotrophic factor (BDNF), neurotrophin-3 (NT-3), and neurotrophin-4 (NT-4). Each NT binds to a specific receptor with high affinity, and NGF binds to tropomyosin receptor kinase A (TrkA), BDNF, NT-4 to tropomyosin receptor kinase B (TrkB), and NT-3 to tropomyosin receptor kinase C (TrkC). NTs also bind with a low affinity to the neurotrophin pan-receptor p75 (p75^NTR^). This binding triggers signal transduction that may lead to cell proliferation or apoptosis, depending on the cellular context and the NT-receptor complex [3]. Moreover, mature NTs are synthesized as large precursor molecules called proneurotrophins, which are cleaved by proteolytic enzymes, such as plasmin [4], furin, and metalloproteinases (MMPs) [5]. These precursor proteins are biologically active and bind with high affinity to the complex formed by p75^NTR^ and its coreceptor, sortilin, inducing apoptosis [3].

As a consequence of their role in cellular proliferation and apoptosis, NT pathways are widely related to the pathology of several types of cancer [6]. Activation of the BDNF/TrkB pathway has been associated with carcinogenic processes, such as invasion, angiogenesis, metastasis, resistance to apoptosis, cell proliferation, drug resistance, and cancer progression [7,8]. Likewise, pro-NGF/NGF pathway activation leads to proliferation and invasion of cancer cells in several types of solid tumors [8]. However, the role of NTs in hematological malignancies has not yet been fully established.

Acute lymphoblastic leukemia (ALL) is the most common cancer in childhood, accounting for approximately 30% of cases in this age group [9]. ALL is a heterogeneous disease with subtypes differing according to cellular and molecular characteristics, response to treatment, and risk of relapse; therefore, it is associated with different outcomes [10]. Fortunately, due to progress in treatment, cure rates have increased extensively over the last few decades [11]. Nonetheless, ALL is still the main cause of death in children and adolescents, primarily in developing countries, where health access barriers are currently present [12]. Thus, studies capable of elucidating the biological mechanisms underlying the disease and its prognostic factors are fundamental for developing new therapeutic approaches and selecting the most suitable treatment.

A study by our research group showed that decreased BDNF levels at leukemia diagnosis in children are associated with a worse prognosis and active disease [13]. Similar outcomes have been identified in neuroblastoma [14] and Wilms’ tumors [15], but the underlying mechanisms remain unknown. The present study aimed to investigate the role of NT pathways in childhood ALL, measure protein levels in pediatric samples, and assess the expression of the main genes involved.

## 2. Materials and Methods

### 2.1. Patients and Biological Samples

This prospective cohort study was conducted between September 2011 and March 2021. Bone marrow (BM) or peripheral blood (PB) samples were obtained from pediatric patients (0–18 years old) diagnosed with acute lymphoid leukemia at the Pediatric Oncology Service of the Porto Alegre Clinical Hospital (n = 97) and Conceição Children Hospital (n = 06). In a previous study by our group [13], BDNF levels were evaluated in paired bone marrow and peripheral blood samples collected from the same subjects at the same disease time point. As no statistically significant differences were found between the two sources, samples from bone marrow and peripheral blood were grouped together for the present analysis.

This study was reviewed and approved by the Research Ethics Committee of both institutions (protocol approval numbers 08511/130023, 2017-0614, 2019-0695, and 22403219.6.3003.5530) and conducted according to the Declaration of Helsinki. Written informed consent was obtained from all participants enrolled in this study according to local regulations for consent for the pediatric population.

Our patient cohort was selected from public referral hospitals providing treatment for childhood cancers. ALL diagnosis was confirmed with bone marrow aspirate altered showing blast level ≥ 20%, followed by histochemical, cytogenetic, immunophenotyping, and molecular tests to determine ALL subtypes. After diagnosis, pediatric patients started treatment with Berlin–Frankfurt–Munich (BFM) family modified protocols. Psychiatric disorders such as autism and schizophrenia were exclusion criteria for study enrollment because NT levels can be altered by these conditions [16]. Biological samples were obtained at diagnosis and other treatment timepoints (induction, maintenance, recurrence diagnosis, recurrence treatment, and follow-up). Patients’ demographic data and clinical information such as leukemia subtype (B-ALL considering preB-ALL and Ph+ samples, T-ALL considering pre-T ALL samples), clinical outcome, BFM risk grade, which refers to the chance that the disease will not respond to treatment or return after an initial response to treatment, remission, relapse, central nervous system (CNS) infiltration, and local of relapse were collected from electronic medical records. For the control group, peripheral blood samples from healthy individuals (HIs) aged 0–18 years (n = 11) were collected (Table 1).

### 2.2. Enzyme-Linked Immunosorbent Assay (ELISA) Assay

To measure the NT levels, 4 mL of PB or BM was collected in Ethylenediaminetetraacetic Acid (EDTA) tubes. Samples were centrifuged at 3000 rpm and the supernatant fraction was frozen at −80 °C. Pro-BDNF, BDNF, and NGF levels were measured using an enzyme immunoassay sandwich kit (pro-BDNF kit number MBS7606020, manufacturer MyBioSource, San Diego, CA, USA, BDNF kit ChemiKine CYT306 manufacturer MilliporeSigma, Burlington, USAand NGF kit number E-EL-H1205 manufacturer Elabscience, Houston, TX, USA) according to the manufacturer’s instructions.

Statistical analysis was performed using the R software version 4.0.2 (22 June 2020) and IBM SPSS Statistics software (version 29.0). Categorical variables were reported as absolute and relative frequencies. Continuous variables were reported as median and interquartile range (IQR 3-1 or 75–25%). Pearson’s Correlation and Spearman’s Correlation were used to compare NT levels, and the Mann–Whitney test was used for nonparametric comparisons between every two groups. Statistical significance was set at *p* < 0.05. To account for multiple testing and control the rate of false positives, the Benjamini–Hochberg false discovery rate (FDR) correction was applied to the *p*-values obtained from differential expression analyses.

### 2.3. Exploratory Overall Survival Analysis

A Kaplan–Meier graph for each NT (NGF, pro-BDNF, and BDNF) was generated using univariate analysis, adopting the median for each protein level at diagnosis as the cut number. A high value was considered if it was higher than the median, and a low value if it was lower than or equal to the median. Log-rank *p*-value was calculated to evaluate the statistical significance using R software, and *p* < 0.05 was considered statistically significant for all analyses. The median follow-up time was calculated from the Kaplan–Meier curve.

### 2.4. Primary Cell Culture

To establish a blast cell culture, 4 mL of BM or PB was obtained from pediatric ALL patients at diagnosis. Blast isolation was performed by centrifugation using Ficoll Paque. The mononuclear cell coat was collected and washed with Phosphate Buffered Saline (PBS). Cells were resuspended in RPMI-1640 medium supplemented with 10% Fetal Bovine Serum, 1% antibiotics, and 0.1% antifungal. The cells were then treated with BDNF 50 ng/mL and maintained at 37 °C with 5% CO_2_. After 72 h of incubation, the cell suspension was homogenized with 0.4% trypan blue 1:1 and counted immediately using a hemocytometer. To establish cell viability after treatment, the control for comparison was determined as the number of cells without treatment, considered as 100% viability.

### 2.5. Microarray Data and Gene Expression Analysis

Gene expression data were obtained from the Gene Expression Omnibus public genomic data repository Gene Expression Omnibus (GEO) [17]. The following pediatric leukemia datasets were analyzed: GSE87070, GSE28460, GSE26713, GSE47051, GSE7440, and GSE101425, all from the same platform (Affymetrix Human Genome U133 Plus 2.0 Array, GPL570). The samples from each dataset were separated into two groups according to their specific clinical characteristics (Table 2). The gene expression of BDNF and NGF was evaluated, as well as ten other genes related to NT pathways, such as NT receptors and converting enzymes—p75^NTR^, TrkB, TrkA, Sortilin-1 (SORT1), Metalloproteinase 2 (MMP-2), Metalloproteinase 3 (MMP-3), Metalloproteinase 7 (MMP-7), Metalloproteinase 9 (MMP-9), Furin (FURIN), and Plasmine (PLG)—in each dataset. Expression analyses were performed using the R software, version 4.0.5, using a series of R/Bioconductor packages. The oligo package version 1.5 [18] was used for sample loading and normalization, whereas the arrayQualityMetrics package 3.42 [19] was used for sample quality control. The Robust Microarray (RMA) method, responsible for background-adjustment, normalization, and log-transformation of expression values present in the oligo package, was performed separately in each dataset [20,21,22]. The Limma package version 3.42 [23] was used for differential expression analysis between groups. Moderated *t*-statistics were employed to evaluate differential gene expression across groups, with false discovery rates controlled using the Benjamini-Hochberg procedure [24]. Genes were considered differentially expressed based on a *p*-value < 0.05.

## 3. Results

### 3.1. NT Levels Are Reduced in ALL Patients at Active Phases of the Disease

This study included 103 pediatric patients with a mean age of 6 years (IQR, 3–9 years) who were diagnosed with ALL. Patients were predominantly male (65%) and diagnosed with B-cell ALL (84%). Remission (assessed on day 33 of initial treatment) and absence of relapse (68% and 74%, respectively) accounted for the majority of patients despite the high-risk classification proportion observed in this cohort (57%) (Table 1).

A total of 204 samples were collected at various timepoints during the clinical course. For the control group, 11 peripheral blood (PB) samples were obtained from healthy pediatric individuals (HIs). The levels of pro-BDNF (1.14 pg/mL), BDNF (0.70 pg/mL), and NGF (1.66 pg/mL) analyzed at ALL diagnosis were significantly lower than pro-BDNF (11.07 pg/mL), BDNF (6.87 pg/mL), and NGF (6.84 pg/mL) levels assessed in HI samples (Figure 1, Figure 2 and Figure 3). pro-BDNF and BDNF levels at treatment induction (pro-BDNF 3.72 pg/mL and BDNF 1.33 pg/mL), relapse diagnosis (pro-BDNF 4.77 pg/mL and BDNF 0.87 pg/mL), and relapse treatment (pro-BDNF 3.86 pg/mL and BDNF 0.32 pg/mL) were also lower when compared to HI (pro-BDNF 11.07 pg/mL and BDNF 6.87 pg/mL) (Figure 1 and Figure 2). For NGF, the levels were lower for these same timepoints (induction = 1.67 pg/mL; relapse diagnosis = 2.43 pg/mL; relapse treatment = 3.41 pg/mL), as well as for maintenance (1.87 pg/mL) and follow-up (2.29 pg/mL) timepoints, when compared to HI samples (6.84 pg/mL), as shown in Figure 3.

When comparing pro-BDNF levels during the ALL timepoints, we observed a decrease at diagnosis (1.14 pg/mL) and an increase during the treatment period at induction (3.72 pg/mL), maintenance (7.73 pg/mL), follow-up (7.09 pg/mL), relapse diagnosis (4.77 pg/mL), and relapse treatment (3.86) (Figure 1). Decreased levels of BDNF were found at diagnosis (0.70 pg/mL), when compared with maintenance (4.50 pg/mL) and follow-up (4.45 pg/mL), as well as decreased levels during disease relapse (0.87 pg/mL) and treatment relapse (0.32 pg/mL) when compared with maintenance (4.50 pg/mL) and follow-up periods (4.45 pg/mL) (Figure 2). No statistically significant difference was observed in NGF during the ALL treatment timepoints (Figure 3). A strong positive correlation (Pearson’s correlation coefficient = 0.87 and Spearman’s correlation coefficient = 0.71) was found between pro-BDNF and BDNF levels at the time points analyzed (Figure 4).

Since NT levels were lower at diagnosis, a comparison between the protein levels at this time point and patient’s disease outcome was performed in order to identify possible prognosis relationships. Analyzing Kaplan–Meier curves within 06 years of median follow-up time (95% CI 5.91–6.10), the death events were increased in patients with reduced pro-BDNF (pLogRank = 0.049) (Figure 5).

### 3.2. BDNF May Reduce Viability of Leukemia Cells

A total of 15 samples from ALL patients at diagnosis were obtained to perform cell culture. Following treatment with BDNF at 50 ng/mL, eight subjects (53%) exhibited reduced cell viability compared to the control group. The observed reduction had a range from 8.11% to 56.25%. In four samples, the viability after treatment was very similar to the control (ranges), and, in three samples, there was an improved cell viability. Statistical analysis was unavailable due to insufficient cells for plating three independent experiments. The samples used for cell culture were not the same as those used to measure NT levels, making it impossible to directly correlate protein levels with culture viability. Additionally, this experiment presents a limitation, as the cultures were performed using primary human cells rather than established cell lines, which typically have greater growth potential and viability. Nevertheless, we chose to include this information in the text, despite the incomplete results, as it represents work conducted during this study and may serve as a foundation for future improvements or replication by other research groups.

### 3.3. BDNF and NGF Expression in ALL Gene Datasets

To explore the role of BDNF and NGF in pediatric ALL, their gene expression was analyzed along with ten related genes in different tissues and/or clinical conditions comprising a total of six datasets (Table 2). A significant difference in *BDNF* expression was found in the GSE87070 dataset. Its expression was higher in the T-ALL compared to the B-ALL samples (Table 3). A similar result was found for *NGF*, with a statistically significant result in the GSE26713 dataset, in which its expression was higher in healthy medullary tissue compared to tumor tissue at diagnosis (Table 3).

### 3.4. NT Receptors Expression in ALL Gene Datasets

*p75^NTR^* was more expressed in B-ALL compared to T-ALL, while the *SORT1* receptor was higher in T-ALL and higher in healthy tissue compared with tumor tissue. When comparing medullary tumor tissue with peripheral blood from individuals with ALL at diagnosis, only the *TrkB* receptor had a significant difference, being more expressed in the bone marrow (Table 3).

### 3.5. Metalloproteinases Gene Expression in ALL Gene Datasets

When analyzing the expression of metalloproteinases, four datasets had positive results. The metalloproteinase *MMP-9* was found to be more highly expressed in relapse compared to diagnosis, and also exhibited greater expression in healthy bone marrow compared to tumor tissue, with the latter result having stronger statistical support (Table 3). The *MMP-3* was found to be more expressed in healthy stroma compared to tumor stroma, and *MMP-2* was more expressed in T cell tumors compared to B cell tumors, a similar result to *BDNF*. *MMP-7* was not found with any significant expression difference in the datasets analyzed. *PLG* and *FURIN* presented differences in expression related to ALL cell types; both were more expressed in B-ALL compared to T-ALL (Table 3).

## 4. Discussion

In order to provide an overview of the NT role in pediatric ALL, the NT protein level and the main genes of NT’s pathways were evaluated. It is known that BDNF and NGF are secreted by bone marrow stromal cells [25], as well as NGF plays an important role in enhancing activation, differentiation, and survival of B and T lymphocytes, monocytes, eosinophils, neutrophils, basophils, and mast cells [1,26]. We observed the decreased levels of the NGF protein in the ALL diagnosis samples when compared to the HI samples. This supports the well-described NGF function in providing the maintenance of immune cells for normal health systems. In acute myeloid leukemia, Simone et al. found higher levels of NGF in patient samples compared to normal controls [27]. To the best of our knowledge, we describe for the first time the under expression of the NGF gene in ALL tissue diagnosis in children compared to healthy bone marrow samples. This is consistent with other studies indicating that circulating NGF levels and NGF gene expression in immune cells are influenced by immunologically related disorders (human inflammatory disease and leukemia) [27,28].

Similarly to NGF, BDNF is also related to B lymphocyte activation [29], maturation [30], and survival [31]. Schuhmann et al. reported that BDNF-deficient mice had a reduced number of B cells when compared with wild-type mice, suggesting that the maturation of B lymphocytes in the bone marrow of deficient mice was blocked at the Pre-BII stage [30]. This suggests a severe influence of BDNF in B-ALL formation, since leukemia causes are intrinsically related to cell development stages. A similar study was conducted by Linker et al., which showed a reduction in the T-cell numbers in peripheral lymphoid organs of BDNF-deficient mice [32]. Our results indicated that BDNF is higher in healthy individuals when compared to patient samples at diagnosis, as well as its precursor pro-BDNF. We also observed that the BDNF gene was more expressed in samples of T-ALL when compared with preB-ALL. This finding is according to the BDNF pathway in maturation steps during the normal B lymphopoiesis. In a recent literature review, Azoulay and Horowitz stated that there was no difference between BDNF levels in patient samples and healthy individuals for other hematological diseases such as mastocytosis, T-cell lymphoma, and myeloproliferative disorders [33]. In leukemia, specifically, BDNF was reported to promote B chronic lymphocytic leukemia cells [34]. These data are specific to chronic leukemia and cannot be correlated with ALL. ALL affects immature cells in the bone marrow during lymphopoiesis (antigen-independent), while chronic leukemia impacts mature cells in the bloodstream (antigen-dependent). In addition to results comparing protein levels in ALL and healthy individuals, here, we also described a clear difference in the NTs’ behavior along ALL timepoints. Pro-BDNF and BDNF levels were decreased at active disease timepoints (diagnosis, induction, and recurrence) compared to treatment maintenance or follow-up, where levels were similar to those in healthy controls. Several studies reported a protective role of BDNF in neurodegenerative disorders [35], cardiac diseases [36], and also a potential role in recovery post-injury [37,38,39]. There is evidence of a relationship between BDNF levels and the severity of diseases, indicating that lower levels of BDNF can be related to poor outcomes [36,40,41,42], while increasing BDNF levels post-injury leads to cell-tissue recovery. In myelosuppressive radiation injury, BDNF promoted faster recovery of mature blood cells and hematopoietic stem cells capable of multi-lineage reconstitution by hematopoietic regeneration [38]. For hematological disease, these data are similar in multiple myeloma (MM), which reported that low BDNF levels were associated with poor prognosis [43]. In chronic lymphocytic leukemia, elevated BDNF levels were related to a favorable outcome [44].

By the date of the publication, it is the first work that evaluates the protein balance between pro-BDNF/BDNF in a hematological disease context. We found a high correlation in the protein levels of pro-BDNF and BDNF within its levels evaluated. Surprisingly, pro-BDNF follows the same pattern as BDNF and is reduced in childhood leukemia samples, discarding the hypothesis of a derangement in the conversion from pro-NTs to mature NTs.

After assessing that NT levels are altered in ALL and it has several influences in leukemogenesis, we decided to investigate an overview profile for the expression of the main genes involved in the NT pathway to better understand why it happens in a disease context. Firstly, the NT receptors are well established as participants in hematopoietic early stages [45,46]. The present work found an NT receptor overexpression in pediatric leukemia samples, in which *p75^NTR^* was overexpressed in B-ALL as well as *SORT1* in T-ALL disease. Proneurotrophins can bind p75^NTR^ in combination with the coreceptor sortilin, inducing apoptosis in neural cells [47]. In cancer, high *p75^NTR^* expression has been shown to be associated with a favorable clinical outcome in adenocarcinoma, lung cancer, and prostate cancer [48,49,50]. In pediatric B-cell precursor ALLhigh *p75^NTR^* expression is a strong prognostic marker able to identify pediatric ALL patients with favorable outcomes [51]. Likewise, the coreceptor sortlin has been investigated for its anti-apoptotic effect that affects the cell death mechanism in leukemia cells [52,53]. Therefore, it is possible to suggest that this effect could happen in similar patterns in pediatric leukemia, although further investigations on the rules of those two receptors in the different types of pediatric leukemias are needed.

In addition to evaluating NT and NT receptors, verifying how the cleavage enzymes are presented in the pediatric disease is also important to understand the role of pro-NTs in this condition. The current study reported that *MMP-9* was more expressed in leukemia relapse samples when compared to diagnosis samples and in healthy bone marrow compared to tumor tissue at diagnosis. Also, *MMP-3* was more expressed in tumor stroma at diagnosis when compared to healthy stroma, and *MMP-2* was more expressed in T cell tumors compared to B cell tumors at diagnosis. Recent studies have highlighted the pathological roles of matrix metalloproteinases in childhood leukemia [54,55] and identified these molecules as key targets for therapeutic advancements in cancer [56]. A higher MMP-9 secretion was related to a lower overall survival (OS) rate in pediatric leukemia [57]. In adult B-ALL, Pan et al. reported similar findings, showing that MMP-9 levels were elevated in high-risk patients as well as in those with extramedullary disease—both of which are associated with poor prognosis [58]. In concordance with our results, MMP-9 may correlate with poor prognosis in leukemia and act as a potential prognostic biomarker. Besides that, the expression of *MMP-2* was also related to extramedullary infiltration in adult ALL, although the same was not verified for childhood leukemia [59]. For MMP-3, there is no data available relating to leukemia, despite its described role in modulating tumor metastasis and progression in several types of cancer [60,61]. In xenograft skin squamous cell carcinoma, the expression of stroma-derived *MMP-3* is exclusively restricted to very late-stage invasive of highly malignant cells transplanted in mice, when massive invading tumor masses are formed, whereas the majority of other MMPs are also expressed in earlier stages [62].

In addition to the metalloproteinases, furin and plasmin convertases are also reported to be involved with cell proliferation, migration, invasion, and vascularization (angiogenesis) in many cancer types [63,64,65,66], although their role in hematological malignancies remains unknown. Here, we described a higher expression of *PLG* and *FURIN* in B-ALL than in T-ALL. There is a lack of evidence that it may be related to the NT maturation process. Further studies are needed to enlighten the role of converse enzymes and NT in pediatric leukemia. To date, this is the first study to investigate the roles of MMPs, furin, and plasmin within the context of neurotrophin (NT) pathways in childhood leukemia.

In addition to investigating the role of NTs in leukemogenesis, we also examine their possible role in unfavorable outcomes in ALL. In our exploratory overall survival analysis using diagnosis protein levels, we found a decreased overall survival probability in 5 years when the pro-BDNF level is decreased, but the same was not found for BDNF. This could be due to the small number of patients and the median follow-up time of our sample. A full survival analysis study, which includes some traditional prognostic factors, like genetic markers and minimal residual disease, is necessary to check if pro-BDNF level at diagnosis is an independent prognostic factor in childhood ALL. Furthermore, the reduction effect in patients’ leukemic cells by BDNF treatment suggests a possible influence of BDNF in response to leukemia therapy, but further studies are required to explore this therapeutic potential. Taken together, these results can suggest that reduced pro-BDNF levels, and possibly BDNF, could be associated with worse clinical stages and a possible poor prognosis in children with ALL.

Despite all efforts to better understand the role of NTs in hematological diseases in the past years, we still have a long way to go. The lack of and controversial data in the literature are a challenge that raises persistence in filling these gaps, particularly in childhood leukemia, the most common cancer in children. The biological characteristics of adult and childhood leukemias differ significantly, as do those of each specific hematological malignancy. Additionally, there is a critical need for global standardization of neurotrophin (NT) measurement methods to minimize biases in biological sample processing and enhance the accuracy and comparability of results worldwide.

## 5. Conclusions

This is the first work that explores all NT pathway members in an in silico and experimental view for pediatric leukemia. At protein levels, NTs are lower at diagnosis and in active timepoints of the disease. We found remarkable differences at the gene level, identifying the main NT pathway genes that are altered in ALL. We also demonstrated a correlation between pro-BDNF and BDNF proteins and a possible poor disease outcome when the pro-BDNF level is decreased at ALL diagnosis. As the primary hypothesis, we expected a poor prognosis when BDNF levels are decreased in diagnosis; however, the small number of samples could affect this result. We also expected to have pro-NGF protein and NT receptor protein (p75^NTR^, sortilin, TRKA, TRKB) levels for the analyzed samples, but these proteins were not assessed in the present study. Besides that, other factors in pediatric ALL, such as genetics, environmental variations, and different treatment regimes, might impact NT levels and their gene expression, but there is no information in the literature to relate them so far. These factors may be analyzed in further studies, with more detailed longitudinal analysis, in order to assess changes in NT levels over time and to completely understand the role of NTs in pediatric ALL.

Our results allow us to suggest that it is the whole context in NT pathways that is one of the causal factors for B-ALL in children. Alterations in expression and protein content of NTs and their respective receptors seem to be related to a maturation arrest and, additionally, an anti-apoptotic effect in leukemia cells. Activation of this pathway through NT-receptor binding disrupts the tumor microenvironment, potentially contributing to poor prognosis and advanced disease stage. Therefore, neurotrophins may serve as valuable biomarkers in the assessment of pediatric acute lymphoblastic leukemia.

## Figures and Tables

**Figure 1 cancers-17-01623-f001:**
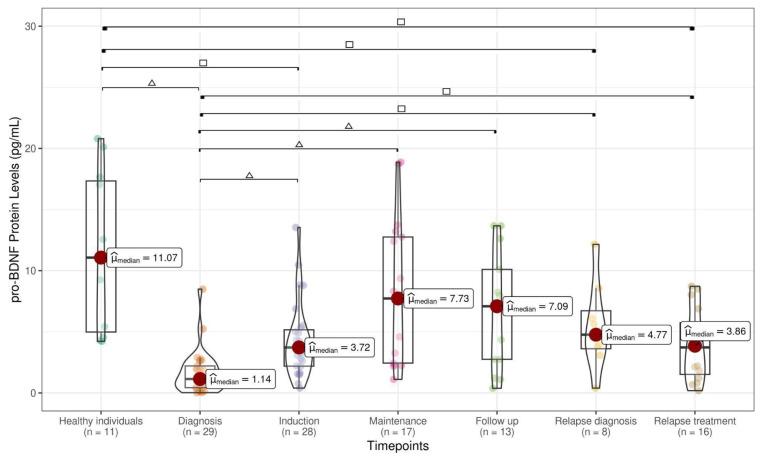
Pro-BDNF levels in healthy individuals and children at ALL timepoints (median IQR 75–25% within the box limits), with statistical significance from the Mann–Whitney test between two groups indicated as △ for *p* < 0.001 and as □ for *p* < 0.05. After FDR correction, all comparisons remained significant except for the comparison between healthy individuals and those diagnosed with relapse, likely due to the small sample size.

**Figure 2 cancers-17-01623-f002:**
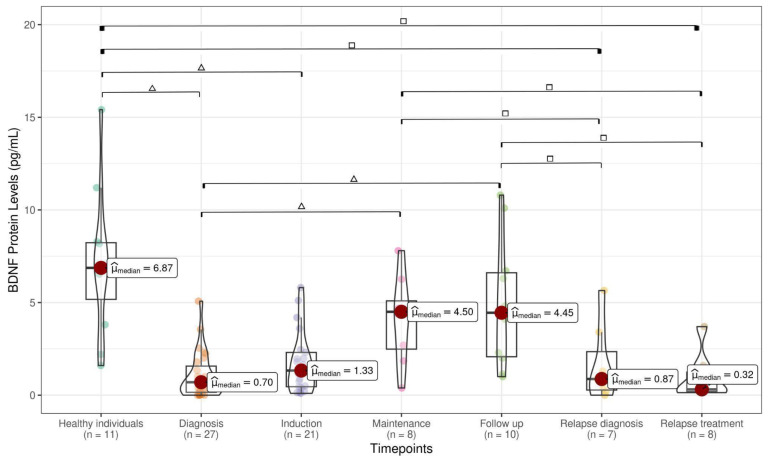
BDNF levels in healthy individuals and children at ALL timepoints (median IQR 75–25% 75–25% within the box limits), with statistical significance from the Mann–Whitney test between two groups indicated as △ for *p* < 0.001 and as □ for *p* < 0.05. After FDR correction, all comparisons remained significant except comparison between disease stages maintenance and relapse diagnosis, likely due to small sample size.

**Figure 3 cancers-17-01623-f003:**
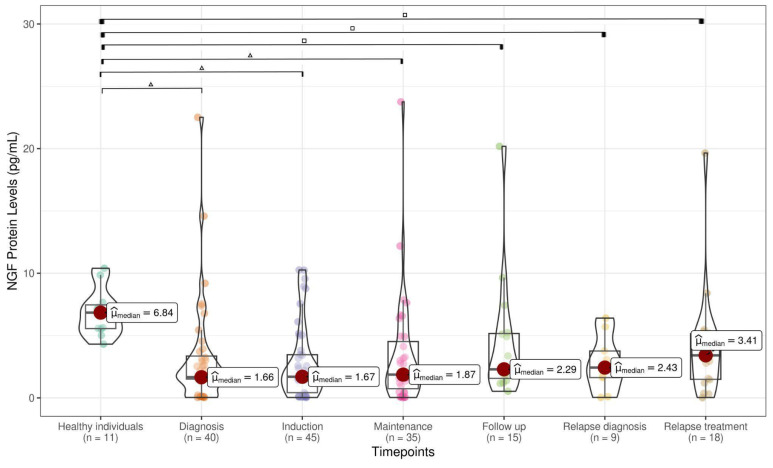
NGF levels in healthy individuals and children ALL timepoints (median IQR 75–25% 75–25% within the box limits), with statistical significance from the Mann–Whitney test between two groups indicated as △ for *p* < 0.001 and as □ for *p* < 0.05. After applying FDR correction for false positives, all previously significant results remained statistically significant.

**Figure 4 cancers-17-01623-f004:**
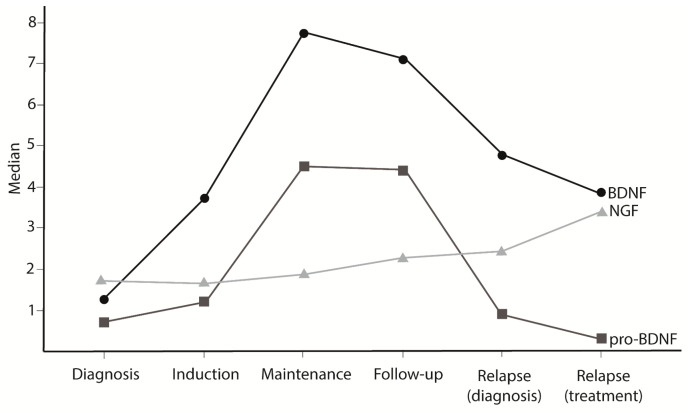
NT levels in ALL patients during clinical evolution. There is a strong positive correlation between BDNF and pro-BDNF levels (Pearson’s r = 0.87). BDNF and pro-BDNF protein levels are lower in diagnosis and relapse when compared to the treatment maintenance and disease remission (follow-up).

**Figure 5 cancers-17-01623-f005:**
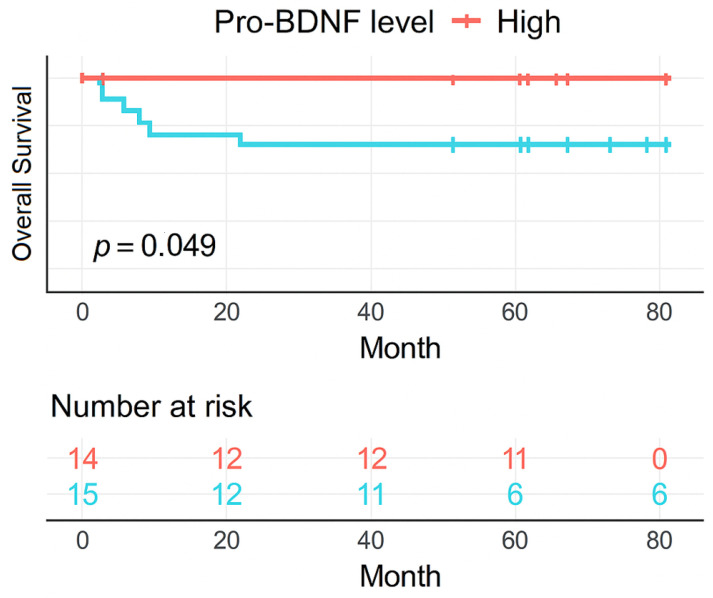
Kaplan–Meier estimate of overall survival (OS) of children and adolescents with ALL stratified by pro-BDNF high and low levels (*p* < 0.05).

**Table 1 cancers-17-01623-t001:** Clinical characteristics of patients included in this study.

Characteristics of Patients (n = 103)	Total (%)
**Gender**	
Male	67 (65)
Female	36 (35)
**Age at diagnosis (years)**	
Minimum	0.5
Maximum	16
Median	6
**ALL Subtype**	
B-ALL *	87 (84)
T-ALL **	16 (16)
**BFM Risk group**	
High	59 (57)
Standard	24 (23)
Intermediary	20 (20)
**Outcome**	
In treatment/remission	78 (76)
Death	25 (24)
**Remission (Day 33 of initial treatment)**	
Yes	70 (68)
No	21 (20)
Remission after relapse treatment	5 (5)
Refractory	7 (7)
**Relapse (Early or late)**	
No	76 (74)
Yes	27 (26)
**CNS infiltration at diagnosis**	
No	91 (88)
Yes	12 (12)
**Local of relapse**	
Bone marrow	13 (42)
Testicle	5 (16)
CNS	5 (16)
CNS and bone marrow	3 (10)
Bone marrow and testicle	2 (7)
CNS, bone marrow and testicle	1 (3)
Eye	1 (3)
Missing	1 (3)

* n B-ALL = 87 including common B-ALL n = 58, pre-B n = 22, pro-B n = 1, Ph+ n = 6; ** n T-ALL = 16 including common T-ALL n = 15 and pre-T ALL n = 1.

**Table 2 cancers-17-01623-t002:** Description of the dataset and comparison between groups.

GSE	Number of Samples	Leukemia Type	Sample Type	Compared Groups
GSE87070	633	B-ALL, T-ALL	Bone marrow	B-ALL (n = 563)/T-ALL (n = 70) At diagnosis
GSE28460	92	B-ALL	Bone marrow	Diagnosis (n = 44)/Recurrence (n = 48)
GSE26713	112	T-ALL	Bone marrow	Diagnosis (n = 106)/HI (n = 6)
GSE47051	97	B-ALL, T-ALL	Bone marrow and peripheral blood	Patient’s diagnosis bone marrow samples (n = 82)/Patient’s diagnosis peripheral blood (n = 15)
GSE7440	80	B-ALL	Bone marrow	Early response to therapy (n = 41)/Slow response to therapy (n = 39)
GSE101425	51	B-ALL	Mesenchymal Stromal Cells (MSCs)	Patients MSCs (n = 35)/ Healthy individuals MSCs (n = 16)

**Table 3 cancers-17-01623-t003:** Significant results of gene expression analysis.

Genes	LogFC	*p*-Value	Status
**B-ALL X T-ALL**			
*BDNF*	−0.171	*p* < 0.001	Overexpressed in T-ALL
*FURIN*	0.148	0.009	Overexpressed in B-ALL
*MMP-2*	−0.114	0.008	Overexpressed in T-ALL
*p75^NTR^*	0.626	*p* < 0.001	Overexpressed in B-ALL
*SORT1*	−0.487	*p* < 0.001	Overexpressed in T-ALL
*PLG*	0.088	0.018	Overexpressed in B-ALL
**Tumor Tissue Diagnosis X Health Tissue**			
*FURIN*	−0.666	0.001	Overexpressed in Healthy Tissue
*SORT1*	−1.129	*p* < 0.001	Overexpressed in Healthy Tissue
*NGF*	−0.274	0.011	Overexpressed in Healthy Tissue
*MMP-9*	−4.902	0.086	Overexpressed in Healthy Tissue
**Diagnosis X Relapse**			
*MMP-9*	−0.819	0.003	Overexpressed in Relapse
**Bone Marrow X Peripheral Blood**			
*TrkB*	0.213	0.002	Overexpressed in Bone Marrow
**Health MSC X Tumor MSC**			
*MMP3*	1.782	*p* < 0.001	Overexpressed in Healthy Bone marrow mesenchymal stromal cells (MSC)

## Data Availability

Data are contained within the article.

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
