# Peer review of "Neurotrophins as Potential Biomarkers for Active Disease and Poor Outcome in Pediatric Acute Lymphoblastic Leukemia"

_cancers, 2025, doi:10.3390/cancers17101623_

Round 1

Reviewer 1 Report

Comments and Suggestions for Authors

Review of:  Neurotrophins as Potential Biomarkers for Active Disease and Poor Outcome in Pediatric Acute Lymphoblastic Leukemia

This team has done a great deal of work on this topic with a publication as far back as 2016.  The current in depth publication generates many unanswered questions on the role of NF in cancer generally and childhood ALL in particular.  This denotes a good study and it is worthy of publication.

The most important question is my mind is “do leukemia cells suppress NF to gain a survival advantage over normal lymphoid cells.  Is this true of both B and T cell leukemia over normal B and T cells?   I look forward to you finding the answer.

Here are my suggestions for improvement

Major

YOU DO MANY STATISTICAL COMPARISONS WHICH CAN RESULT IN FALSE POSITIVE RESULTS.   YOU SHOULD ACKNOWLEDGE THIS IN YOUR DISCUSSION.

Detecting and avoiding likely false-positive findings – a practical guide

Wolfgang Forstmeier, Eric-Jan Wagenmakers, Timothy H. Parker

First published: 23 November 2016 https://doi.org/10.1111/brv.12315

WHAT WERE THE CAUSES OF DEATH?  WERE THEY ALL DUE TO RELAPSE OR TREATMENT OF RELAPSE?

I ASSUME YOU USED DEATH IN YOUR KM SURVIVAL CURVES AS RELAPSE WAS NOT SIGNIFICANT DUE TO SMALL NUMBERS.   PLEASE CONFIRM

“Psychiatric disorders such as autism and schizophrenia were 99  exclusion criteria for study enrollment because NTs levels can be altered by these 100  conditions [16]”

HOW WERE THE DIAGNOSES FOR EXCLUSION MADE AND HOW MANY WERE EXCLUDED

“For the control group, peripheral blood samples from healthy 108  individuals (HI) aged 0-18 years old (n=11) were collected”

THIS IS A RELATIVELY SMALL SAMPLE.  I NOTED YOU HAD MANY MORE HI IN YOUR 2016 PUBLICATION.  WHY COULD YOU NOT ADD THESE TO YOUR HEALTHY CONTROLS?

LINE 133 TYPO  “P-logHank”

Remission (assessed on day 33 of initial treatment) and absence of relapse (67 % and 73.8 %, respectively) accounted for the majority of patients despite the high-risk classification proportion observed in this cohort (57 %) (Table 1). 

THIS IS UNCLEAR.  DO THE PERCENTAGES APPLY TO TWO DIFFERENT RISK GROUPS?  PLEASE CLARIFY

Minor

  1. Significant figure.   You use four and to declutter I would suggest 2.  For example. 34.95% change to 35%

2) Use use myelogram which I assume is what you call bone marrow.  I would change to bone marrow for us Anglos

3) You use a comma in some numbers which I assume means a period.  I would use a period

4) Figure 1. pro-BDNF levels in healthy individuals and children ALL timepoints (median 196  IQR 75%-25%), statistical significance from Man Whitney test between each 2 groups 197  reported as â–³ if p<0.001 and as  if p<0.05.

PLEASE STATE WITH THE CAPTION THAT THE LINES REPRESENT UPPER AND LOWER VALUES (OR ARE THEY PERCENTILES)?

5) TABLE 3. CHANGE “HEALTH TISSUE” TO “HEALTHY TISSUE”

Comments on the Quality of English Language

given in comments

Author Response

Please see document attached.

Reviewer 2 Report

Comments and Suggestions for Authors

de Andrade and colleagues reported findings from measurements of blood and/or bone marrow pro-brain-derived neurotrophic factor (pro-BDNF), BDNF and nerve growth factor (NGF) levels in pediatric ALL patients.  Authors may wish to address several specific comments and concerns to further improve the manuscript.

Specific comments:

1. Authors mentioned a total of 204 samples collected from 103 ALL patients at different stages of disease progression/treatment. Authors should disclose the information as how many of these are blood (plasma) samples and how many are bone marrow (plasma) samples. Authors should also give brief description as how the 4 mL bone marrow cells was collected/processed from each patient. Authors need to make a statement, with the backup of some data analyses, indicating that readout from blood and BM were similar such that authors could use them indiscriminately in the data analyses.  

  1. Data presented in Figures 1, 2, and 3 should be from the same set of 204 samples collected from 103 ALL patients at various stages. However, the number of observations differed among the figures for each sub-group! Why? The number of observations in each sub-group should be the same for each of the three measurements! And more importantly, the number of observations of all sub-groups should add up to 204 in each figure! Currently that is not the case. Where are the missing observations and why they are not included? Authors need to present all data without bias.
  2. Line 237 pointed to Figure 6 for the presentation of data from cell culture study. However there was no Figure 6 in the current version of the manuscript. Authors need to present data in Figure 6. Since authors claimed that BDNF reduces viability of leukemia cells (line 233), authors need to provide statistical conformation to support this claim.
  3. Authors reanalyzed gene expression data published by others but description of the reanalyzed data, as detailed between lines 242 and 262, did not make a good connection with data observed from current study. Authors should make an effort to related the gene expression data, especially the expressions of pro-BDNT, BDNT and NGF under various conditions, to strengthen or weaken the conclusion authors intend to make based on current data.  Data presented in current Table 3 does not make too much sense and need to be reformatted such that the data would appear more meaningful to readers.
  4. Each abbreviation needs to be defined at its first appearance in the abstract as well as in the main text before it is used consistently throughout the manuscript.  

Author Response

Please see document attached.

Round 2

Reviewer 2 Report

Comments and Suggestions for Authors

Manuscript was somewhat improved. Authors should add a sentence in method, pointed to reference 13, to indicate that data from blood and bone marrow plasma analyses were similar based on a previous study. 

None assumed this is a longitudinal study therefore no need to add that in the text. While authors did not directly respond to some of the specific review comments, authors' rebuttal could be considered acceptable for this manuscript, understanding the difficulty/complexity in managing clinical sample collection/analyses.     

Author Response

Manuscript was somewhat improved. Authors should add a sentence in method, pointed to reference 13, to indicate that data from blood and bone marrow plasma analyses were similar based on a previous study. 

Author: A sentence was added, thank you for this considerations. It is highlighted in red in the word document attached.

None assumed this is a longitudinal study therefore no need to add that in the text. While authors did not directly respond to some of the specific review comments, authors' rebuttal could be considered acceptable for this manuscript, understanding the difficulty/complexity in managing clinical sample collection/analyses.     

Author: Thanks for the consideration, sentence about longitudinal study was removed from the text.